# Multiplex nodal modularity: A novel network metric for the regional analysis of amnestic mild cognitive impairment during a working memory binding task

Avalon Campbell-Cousins[1]*, Federica Guazzo[2], Mark E. Bastin[3], Mario A. Parra[4], Javier Escudero[1]

1 School of Engineering, Institute for Imaging, Data and Communications, University of Edinburgh, Edinburgh, Scotland, United Kingdom, 2 Human Cognitive Neuroscience, Psychology, University of Edinburgh, Edinburgh, Scotland, United Kingdom, 3 Centre for Clinical Brain Sciences, University of Edinburgh, Edinburgh, Scotland, United Kingdom, 4 Department of Psychology, University of Strathclyde, Glasgow, Scotland, United Kingdom

* Avalon.Campbell-Cousins@ed.ac.uk; Javier.Escudero@ed.ac.uk

**Data availability statement:** MCI data part of the previously mentioned study

## Abstract

Modularity is a well-established concept for assessing community structures in various single and multi-layer networks, including those in biological and social domains. Brain networks are known to exhibit community structure at a variety of scales—local, meso, and global scale. However, modularity, while useful in describing mesoscale brain organization, is limited as a metric to a global scale describing the overall strength of community structure. This approach, while valuable, overlooks important variations in community structure at node level. To address this limitation, we extended modularity to individual nodes. This novel measure of nodal modularity ($nQ$) captures both mesoscale and local-scale changes in modularity. We hypothesized that $nQ$ would illuminate granular changes in the brain due to diseases such as Alzheimer's disease (AD), which are known to disrupt the brain's modular structure. We explored $nQ$ in multiplex networks of a visual short-term memory binding task in fMRI and DTI data in the early stages of AD. While limited by sample size, changes in $nQ$ for individual regions of interest (ROIs) in our fMRI networks were predominantly observed in visual, limbic, and paralimbic systems in the brain, aligning with known AD trajectories and linked to amyloid-$\beta$ and tau deposition. Furthermore, observed changes in white-matter microstructure in our DTI networks in parietal and frontal regions may compliment studies of white-matter integrity in poor memory binders. Additionally, $nQ$ clearly differentiated MCI from MCI converters indicating that $nQ$ may be sensitive to this key turning point of AD. Our findings demonstrate the utility of $nQ$ as a measure of localized group structure, providing novel insights into task and disease-related variability at the node level. Given the widespread application of modularity as a global measure, $nQ$ represents a significant advancement, providing a granular measure of network organization applicable to a wide range of disciplines.

(DOI: 10.1186/s13195-022-01082-9) cannot be shared publicly without clinical research access approval from NHS Lothian and cannot be shared with any 3rd party as per their confidentiality and disclosure of information policy. Those who are interested in working with this data can do so by contacting the University of Strathclyde's ethics committee (ethics@strath.ac.uk) or Prof. Mario A. Parra who manages the data and will assist in applying for clinical research access. All other datasets used in the manuscript are publicly available. For instance, The NKI-Rockland dataset is publicly available with access, study information and pre-processing details given here doi: 10.3389/FNINF.2012.00028. Additionally, Zachary's Karate Club can be accessed at http://konect.cc/networks/ucidata-zachary/.

**Funding:** ACC is funded by the Principal's Career Development Scholarship (PCDS - https://institute-academic-development. ed.ac.uk/postgraduate/doctoral/career-management/principals-scholarships) for his PHD from the University of Edinburgh. MAP received funding from the Alzheimer's Society (https://www.alzheimers.org.uk/) towards the Longitudinal Study of MCI through the Grants AS-R42303 and AS-SF-14-008. This funded the collection of MCI data (used in the manuscript) and specifically in conjunction with this research article 10.1186/s13195-022-01082-9. The funders had no role in study design, data collection and analysis, decision to publish, or preparation of the manuscript.

**Competing interests:** The authors have declared that no competing interests exist.

## Introduction

Alzheimer's disease (AD) and several other dementias often develop from Mild Cognitive Impairment (MCI) [1,2]. MCI is defined as a cognitive decline which is greater than that caused by normal aging [1]. While not everyone with MCI goes on to develop dementia, those with MCI are at much greater risk. MCI diagnosis is typically achieved through a collection of cognitive questionnaires, screening tests, and neuropyschological examinations [3], which are used to benchmark and assess changes in memory, visuo-spatial ability, language, and behaviour. For individuals presenting with the amnestic (memory-related) subtype of MCI (aMCI), this progression frequently results in AD, thereby leading aMCI to often be considered a prodromal stage of the disease [2]. Those who have a diagnosis of aMCI and who convert to AD (MCI converters) typically do so at a conversion rate of 12% per year [4].

Recent studies of AD biomarkers have presented significant promise in improving diagnosis and our understanding of disease progression. However, while the combination of biological biomarkers, neuropsychological tests, and genetic risk markers can allow us to pinpoint the progression of MCI to AD with high accuracy [5,6], methods with higher specificity that explore how individual regions drive this change are needed. This motivates the development of novel biomarkers for AD, especially those sensitive to the early-stages of disease, where less damage has been done.

One such accepted neurocognitive biomarker is the Visual Short-Term Memory Binding Task (VSTMBT), introduced by Parra et al. [7]. It is a task sensitive to early changes in AD, and is composed of eight non-nameable shapes and colours with three phases – encoding, maintenance, and probe. The VSTMBT targets visual memory binding in the brain, a process responsible for the temporary retention of complex objects (i.e., coloured shapes). In addition, tasks which target short-term memory binding are especially important for AD given that they remain relatively unchanged with age while being highly sensitive to the disease [7]. This sensitivity is specific to AD where conjunctive short-term memory binding (i.e., between colour and shape) is impaired as opposed to other, non-AD, dementias where this effect is not observed [8]. This evidence suggests that deficits in short-term memory binding are a preclinical marker of AD.

Using neuroimaging methods such as functional magnetic resonance imaging (fMRI) or electroencephalography (EEG), the dynamics in brain activity during a cognitive task, such as the VSTMBT, can be measured. As AD progresses, changes in functional brain activity have been associated with lower brain efficiency and reduced functional connectivity between brain sub-networks, leading to the naming of AD as a disconnection syndrome [9–11]. This has been further explored in measures of white-matter density and microstructure, measured using Diffusion Tensor Imaging (DTI) [12]. For instance, in [10], white matter structures in the frontal and temporal lobes are found to be vulnerable in early-stage damage caused by familial AD with associated impairments in memory-binding.

To explore these changes, it is natural to model the brain as a network. Networks typically model brain regions as nodes and the connections (edges) between them encode information on the relationship between those regions' function (functional connectivity) or structure (structural connectivity) [13]. In the case of fMRI, edges are often a measure of functional co-activation between blood oxygen level dependent (BOLD) time-series of a pair of brain regions [14]. Meanwhile, in DTI, an edge typically represents the fractional anisotropy (FA) between regions (measuring white matter microstructure), or streamline density (white matter density) [15]. These models capture brain topology, revealing fundamental insights into how the brain is functionally and structurally organized, and how this changes due to disease.

More recently, this approach has been extended to multiplex networks in order to capture additional complexity. Multiplex networks, a type of multi-layer network, consist of multiple single-layer networks (each with the same number of nodes) where the connections within each layer describe a different type of interaction [16]. For instance, multiplex brain networks can be modelled with each layer representing windows of time in an fMRI scan or individual frequency bands in EEG [17]. In this way, multiplex networks have revealed insights into how the brain reorganizes itself in time during a learning task [18], model the cross-frequency dynamics of brain networks [19], and explore its structure-function relationship [20].

This type of modelling is important as brain networks are not random but organized and efficient at both local and global scales [21–23]. To capture the complex interactions between brain regions, their topology, and how these networks dynamically change and reorganize in time, community detection algorithms are employed [16]. The aim of such approaches (such as modularity maximization) is to segregate the network into communities or modules (groups of nodes), where connections within modules are more dense, to describe the underlying organization of the system. Modularity has been explored extensively for many single-layer biological networks, and in how these evolve and adapt due to age or disease [24]. However, the extension of modularity from single to multiplex networks was only achieved recently, and thus has seen less study [25]. For single-layer networks, modularity has been shown to increase along the disease spectrum of AD [26]. This resting-state fMRI (rs-fMRI) study found that changes in key network metrics (prominently modularity) indicated a reduced ability to integrate information distributed across brain regions as a result of AD. In addition, modularity was highlighted as a more sensitive network measure to MCI and AD than other more frequently used measures such as clustering coefficient and path length [26].

A limitation of modularity in single and multiplex networks is that it has been exclusively studied at a *global* scale. It is not fully understood how individual regions of a network, such as the brain, change in modularity due to disease, cognitive task phase, or dynamically change in time. Parra et al. showed that not only did the VSTMBT require specific memory binding regions, but also interacting and/or overlapped brain Regions of Interest (ROIs) for object recognition [27]. While global measures of modularity could allow us insight into the overarching structure of the VSTMBT, it is the extent that these individual ROIs interact and work together that is not well understood. As such, we hypothesized that a novel extension of modularity to individual ROIs, and applied to a multiplex framework, would yield novel insights into the regional modularity of the VSTMBT and how an ROI's contribution to modularity changes as a result of AD. This code has been made publicly available at https://github.com/AvalonC-C/Nodal_Modularity.

## Materials and methods

### Participants

Participants were recruited from the Psychology Volunteer Panel at the University of Edinburgh, volunteers from the Scottish Dementia Clinical Research Network interest register, and referrals by old age psychiatrists based at the NHS Lothian and NHS Forth Valley. Eligibility followed from a variety of criteria such as an age over 55, no neurological or psychiatric diseases effecting cognitive function, and normal or corrected to normal vision [28].

MCI patients had to demonstrate the capacity to consent, were provided with an information sheet informing participants to the longitudinal nature and assessments involved, and signed a consent form prior to involvement in the study. Approval was obtained from the NHS Multi-Site Research Ethics Committee (reference number 06/MRE07/40) and was given

approval by local NHS R&D offices (Lothian R&D: 2006/P/PSY/22 and Forth Valley: FV682). Please see additional details in [28]. Furthermore, access to this dataset for the research described in this paper was obtained from NHS Lothian under study number 2006/P/PSY/22 on 17/11/2021.

This longitudinal study assessed participants with a battery of neuropsychological tests commonly used to asses dementia, such as the Addenbrooke's Cognitive Examination Revised (ACE-R) and the Hopkins Verbal Learning Test Immediate Total and Delayed Recall, and a novel VSTMBT, grouping subjects into early Mild Cognitive Impairment (early MCI), MCI, and those who converted to Alzheimer's disease after a 2-year follow up (MCI converters). Further information on this dataset is available here [28].

From these subjects, a subset underwent fMRI (during which they performed the VSTMBT) and diffusion MRI (dMRI) scanning. Refer to Table 1 for those who met this criteria and passed pre-processing requirements detailed in the following fMRI and DTI sections. For insight into the neuropsychological tests completed and a statistical analysis of these variables between disease groups see S3 Table.

## Visual short-term memory binding task

Two tasks were explored in our study using non-nameable shapes and non-nameable colours derived by Parra et al. [7]. We refer to the first task as *shape*, where only the shapes are presented to the subject. The second we call *binding*, where coloured shapes are presented to the subject. In both cases, the experimental procedure follows as in Fig 1.

During the probe phase of the procedure, subjects would click a button in either their left or right hand indicating whether or not they believed that the new set of shapes/coloured shapes were different or the same (in 50% of the trials the new set of shapes/coloured shapes would be different). Trials in which the subjects made an incorrect choice were omitted from our analysis as they could stem from lack of engagement or attention. Correct trials were more consistent in capturing the underlying cognitive processes of the VSTMBT due to greater participation with the task itself.

## DTI

dMRI data were collected at the University of Edinburgh Brain Research Imaging Centre through a GE Signa Horizon HDxt 1.5T clinical scanner: 3 $T_2$-weighted (b = 0s mm$^{-2}$) and sets of diffusion-weighted ($b$ = 1000s mm$^{-2}$) single-shot spin echo-planar (EP) volumes acquired with diffusion gradients applied in 32 non-collinear directions. Subsequent volumes were acquired in the axial plane (fov = 240 × 240mm; matrix = 128 × 128; thickness = 2.5mm), giving voxel dimensions of 1.875 × 1.875 × 2.5mm. The repetition and echo times were 13.75s

**Table 1. Demographic variables of MCI patients and healthy controls at initial screening. After a 2-year follow up, 6 MCI subjects had converted to AD.**

|  | MCI | early MCI | Healthy Controls |
|---|---|---|---|
|  | N = 16 | N = 7 | N = 8 |
|  | $M \pm SD$ | $M \pm SD$ | $M \pm SD$ |
| **Age** | 74.81 ± 6.09 | 79.71 ± 5.82 | 79.5 ± 5.15 |
| **Years of Education** | 12.81 ± 3.54 | 16.57 ± 3.91 | 15.0 ± 3.54 |
| **Sex** | 9 men; 7 women | 5 men; 2 women | 2 men; 6 women |

Note: N = Number of subjects; M = Mean; SD = Standard deviation.
Anova results on Age: F(2,28) = 2.62, p = .09, $\eta^2$p = .15.
Anova results on YoE: F(2,28) = 2.86, p = .07, $\eta^2$p = .17.

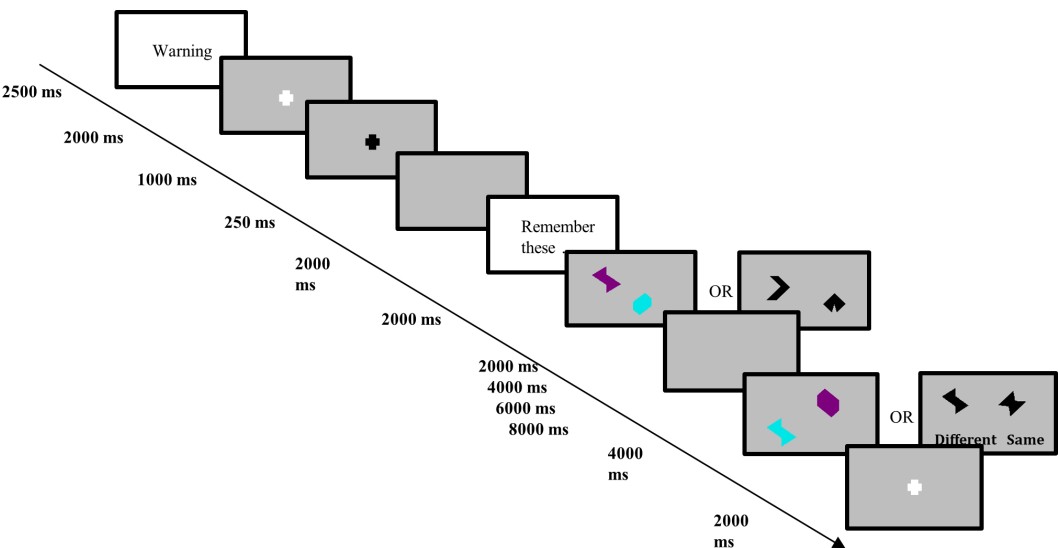

**Fig 1. Task procedure.** Trials were conducted as follows. A warning screen for 2500ms, a fixation period of 3000ms where a white cross turns from white to black, a blank grey screen for 250ms, a reminder of the instructions for 2000ms, the shapes or shapes with colours (depending on shape or binding task) are displayed for 2000ms (encoding phase), a blank grey screen is displayed for a variable time of (2000, 4000, 6000, or 8000ms) due to the fMRI design optimisation [27] (maintenance phase), a second set of shapes or shapes with colours are displayed for 4000ms (probe phase), and lastly an inter-trial interval of 2000ms. Repeat.

and 78.4ms, respectively. A $T_1$-weighted inversion-recovery prepared fast spoiled gradient-echo (FSPGR) volume was also acquired in the coronal plane with 160 contiguous slices and 1.3 mm$^3$ voxel dimensions.

This volume was parcellated into 85 ROIs with the Desikan-Killiany atlas with additional regions acquired via sub-cortical segmentation: accumbes area, amygdala, caudate, hippocampus, pallidum, putamen, thalamus, ventral diencephalon, and the brainstem. This volumetric segmentation and cortical reconstruction was performed with FreeSurfer v5.3.0 using default parameters. For further pre-processing detail please refer to Buchanan et al. [29].

Briefly, pre-processing was conducted with the FSL v6.0.1 toolkit. dMRI data underwent eddy current correction, diffusion tensors were fitted at each voxel and FA was estimated, skull stripping and brain extraction were performed, cross-modal non-linear registration was used to align neuroanatomical ROIs to diffusion space, and the tractography was based on the probabilistic method and white matter seeding approach as in [29].

Of note, the DTI weighted networks were constructed using the streamlines connecting each pair of the 85 grey matter ROIs. The weights of these edges were determined using the mean FA along the interconnecting streamlines [29].

## Task-fMRI

fMRI data was collected during the same appointment, acquisition protocol and with the scanner outlined in the prior DTI section. Once localisation scanning was completed, a structural T1 weighted sequence was acquired (5 contiguous 5 mm coronal slices; matrix $= 256 \times 160$; fov = 240mm; flip angle 8°). During the VSTMBT, contiguous interleaved axial gradient EPI were collected alongside the intercommissural plane throughout

two continuous runs (TR/TE = 2000/40ms; matrix = 64 × 64; fov = 240mm; 27 slices per volume; thickness = 3.5mm; gap = 1.5mm). This yielded 9-minutes of scan in total, comprising 534 volumes (the first three volumes were discarded at the start of shape and binding trials). For clarity, this resulted in 267 volumes for each of the shape and binding tasks per subject.

Using SPM12, fMRI pre-processing follows as in [27]. Outlier detection was used to detect slices with a variance greater than 5 standard deviations [30]. Outlier slices were replaced by an averaged image of the previous and consecutive scans. These images were removed when constructing the network (0.75% of total scans). To account for movement, realignment of each fMRI image to the mean volume of the scan session through B-spline interpolation was done. Slice-timing correction was completed to account for differences in time when acquiring each voxel signal (temporal sync interpolation). Images were then co-registered to their structural $T_1$ images. Lastly, normalization to the MNI space was conducted using segmentation parameters and DARTEL diffeomorphic mapping functions [31,32]. fMRI images were also visually inspected for noise and artefacts and subjects who did not pass inspection were removed from the study.

### Functional network construction

We use the modified Desikan atlas obtained for each subject (as in the DTI section) to define ROIs and resampled to fit the voxel dimensions of the fMRI data. This was done with SPM12 using nearest-neighbour interpolation to ensure that voxel resizing maintained correct ROI mapping. These atlases were also visually inspected to check for proper fitting. For each ROI, the mean signal time-series was extracted from the fMRI images by taking the average signal across voxels defined by the ROI. This was repeated for each image and ROI across the 9-minute scan resulting in time-series of average brain activity at each of our 85 ROIs.

Next, we apply a 0.06Hz high-pass filter to the signals with a sampling rate of 0.5Hz to match our TR of 2s. This was done to account for fMRI signal drift in the very low frequency range [33], and the choice of 0.06Hz relates to the lowest frequency occurring for our longest task trial of 16s. We chose not to low-pass our signal given that, in some cases, our encoding/maintenance task phase is very fast, and thereby in close proximity to the signal's Nyquist frequency.

We define two task stages for our network construction – 'encmaint' and probe. We define encmaint as a combination of the encoding and maintenance phases of the task described earlier in Fig 1. Due to an encoding phase and TR of 2s, this combination of the two phases improves our construction in two ways. It improves our measure of correlation between brain regions for network construction (higher number of samples) and additionally allows us to capture the peak of the haemodynamic response function (HRF) that occurs approximately 5s after stimuli onset (further details in S1 Appendix). The probe phase, described in Fig 1, is shifted forward 2s to decrease the overlap between encmaint and probe phases, improve the capture of peak HRF, while introducing minimal noise from the following inter trial interval.

After defining the task phase windows for each task (binding and shape), we reconstruct our time-series from the repetitions of task phases across the 31 trials. More specifically, the time-series for the probe phase is assembled from the windows corresponding to each probe phase within a trial, in sequential order. The same is done for the encmaint phase. From these time-series, we construct an 85×85 connectivity matrix for each task phase by calculating the Spearman correlation between each ROI pair.

Typically, studies of fMRI brain networks use Pearson correlation in the construction of connectivity matrices. However, it is expected that some noise and outliers may remain in fMRI data. These factors, along with additional pre-processing, task windows which could

not be constructed perfectly due to inherent limitations of fMRI or the task itself, and other confounding factors such as those present due to the calculation of the mean time-series of regional data, made us select the more conservative Spearman's correlation for our analysis due to its robustness to outliers [34], and suitability for non-normally distributed data [34,35]. We also acknowledge a recent study which discovered brain wide increases in functional connectivity with fMRI scan duration [36]. However, we expect this effect to be minor given that our analysis focuses on comparisons between subject groups (where this effect is ubiquitous) and reduced by randomized presentations of shape and binding tasks over the course of the scan.

To model the dynamics between the two task windows, we construct a dual-layer matrix from each of the two encmaint and probe matrices as in Fig 2a. This is done by connecting each node in one layer to its spatial replica in the other via an edge. These edges are weighted equally and set to 1 as is standard in prior studies on multiplex brain networks [18,20,37]. The choice to use weighted graphs, rather than binary, was due to the preservation of the strength of functional association which is complementary to network organization [38].

## Nodal modularity

Formally, we define $G = (V, E)$, as a *weighted undirected graph* composed of a finite set of *vertices*, $V = \{v_1, v_2, ..., v_n\}$, and a two element subset of $V$ called *edges* ($E \subset (v_i, v_j) | v_i, v_j \in V\}$) [39]. When a pair of vertices share a single edge $(v_i, v_j) \in E$, we say that the two vertices are *adjacent* to each other and that the edge is *incident* to each of our vertices. In addition, we define the *adjacency matrix*, $A$ of our graph $G$, as the symmetric $(n \times n)$ matrix of all vertex pairs where entries $A_{ij} \neq 0$ if $(v_i, v_j) \in E$ and $A_{ij} = 0$ otherwise. Furthermore, we define a *multi-layer network* as a family of graphs $G_s = (X_s, E_s)$ (in our case weighted and undirected), also called layers, with $E = \{E_{sr} \subseteq X_s \times X_r; s, r \in \{1, ..., M\}, s \neq r\}$ as the set of connections

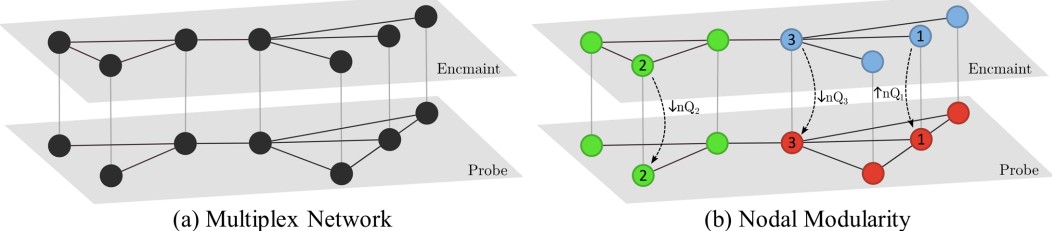

|           (a) Multiplex Network           |           (b) Nodal Modularity           |

**Fig 2. Multiplex network of the VSTMBT and *nQ* example.** a) Networks for the two task phases of the VSTMBT, encmaint and probe, are constructed from the functional co-activations (correlation in time-series) between all pairs of ROIs. Spatial replicas are connected via an inter-layer edge, as seen in the above figure (light grey edges between the two layers), allowing for continuity of network topology in time. b) Here, multiplex nodal modularity (*nQ*) has been calculated for each node in our network as in Eq (1) (refer to the following section). Each colour represents a separate module (obtained from standard multiplex modularity [25], where modules can exist within a layer (blue and red nodes) or across layers (green nodes). We note that values of *nQ* are influenced by, but not entirely dependent on, layer and module assignment. For instance, node 2 undergoes a decrease in *nQ* from the encmaint to probe layer due to a change in connectivity despite no change in module (green). On the other hand, node 1 undergoes an increase in *nQ* from the encmaint to probe layer due to increased connectivity within its module while also undergoing a change in module assignment (blue to green). While both node 1 and 2's changes in modularity are largely driven by changes in connectivity tied to that node, to illustrate how changes in module assignment influence *nQ* consider node 3. Node 3 undergoes no change in connectivity. However, it observes a decrease in *nQ* due to an increase in connectivity of node 1. While modularity in general has increased ($nQ_{1(blue)} + nQ_{3(blue)} < nQ_{1(red)} + nQ_{3(red)}$), the role of node 3 within its module has decreased. It is this interplay between connectivity, modules, and how these change across the layers of the network that influence each node's nodal contribution to classical multiplex modularity.

between nodes of different layers $G_s$ and $G_r$ with $s \neq r$. The elements of each $E_s$ are called the *intra-layer* (within layer) connections and the elements of each $E_{sr}$ ($s \neq r$) are called the *inter-layer* (between layer) connections [40]. A *multiplex network* is a type of multi-layer network in which $X_1 = X_2 = ... = X_M = X$ (all layers have an equal number of nodes) and the only inter-layer connections occur among node replicas, i.e., $E_{sr} = \{(x, x); x \in X\}$ for every $s, r \in \{1, ..., M\}, s \neq r$. As a reminder, for this study, the entries of $A_{ij}$ are defined by the correlation in fMRI signal time-series between all pairs of brain regions in the case of our functional networks. For our structural networks, entries of $A_{ij}$ are defined by the mean FA of white matter between all pairs of brain regions.

In network neuroscience, community detection algorithms are often used to obtain a global measure of how modular the network is, how many modules there are, and in multiplex cases, the flexibility of each node (how often a node switches community through the layers of a network) [23]. However, aside from flexibility, the measure of modularity lacks granularity. The use of modularity as a global measure, while useful as a marker of whole brain changes due to disease, fails to capture to what extent these modular subsystems change in time, due to disease, or otherwise.

To tackle this, we extend the standard multislice (multiplex) modularity quality function, $Q_{multislice}$ [25], to individual nodes as in [41]:

$$Q_i = \frac{1}{2\mu} \sum_{jsr} \left[ \left( A_{ijs} - \gamma_s \frac{k_{is}k_{js}}{2m_s} \delta_{sr} \right) + \delta_{ij}C_{jsr} \right] \delta(g_{is}, g_{jr}) \tag{1}$$

More specifically, we use the multiplex version of modularity (refer to [25]) to calculate the optimized group assignment, $g$, a priori. It follows that, for a node $v_i$ on our graph $G$, its nodal contribution to modularity ($Q_i$) is defined as follows. For all edges of node $v_i$ that exist within the same group ($\delta(g_{is}, g_{jr} = 1)$ when $g_{is} = g_{jr}$) on layers $s$ and $r$, we have the sum over all edge weights ($A_{ijs}$) minus the expected edge weight if edges were placed at random ($\frac{k_{is}k_{js}}{2m_s}$). Here, $k_{is}$ denotes the degree of $v_i$ (the sum of edge weights incident to $v_i$) and $m_s$ is the sum of all edge weights on layer $s$. $\gamma_s$ is the standard intra-layer resolution parameter, which can modify the resolution of communities within each layer. In this paper, we set this to the default value of 1 [25] for simplicity. $\delta_{sr}$ and $\delta_{ij}$ facilitate the calculation of intra-layer and inter-layer edges separately. Thus, $(A_{ijs} - \gamma_s \frac{k_{is}k_{js}}{2m_s} \delta_{sr})$ describes the multiplex version of the observed number of edges minus the expected number for intra-layer edges on each layer. In the case of inter-layer edges, these are handled by $C_{jsr}$ (inter-layer coupling parameter) which is the weight of a node connected to itself across layers s and r. Typically, all inter-layer edges are equal and set to a default value of 1. The inter and intra-layer weights are included in the sum when the nodes share the same group assignment (groups can exist both within and across layers), $\delta(g_{is}, g_{jr})$, and multiplied by $\frac{1}{2\mu}$ where $2\mu = \sum_{jr} k_{jr}$ which comes from the steady-state probability distribution used to obtain the multislice null model, detailed here [25].

The modularity at each node, $Q_i$, can then be calculated using Eq (1) with $g$ as an optimized group assignment, defined a priori and discussed in the following section. In this way, we calculate each node's contribution to global multiplex modularity as

$$Q_{multislice} = \sum_i Q_i. \tag{2}$$

From now on, we refer to nodal modularity as $nQ = Q_i$, and note that $nQ$ can just as easily be defined for single-layer networks. In this case, the expression for multiplex $Q_i$ in

Eq (1) reduces to

$$Q_i = \frac{1}{2m} \sum_j \left[ A_{ij} - \frac{k_i k_j}{2m} \right] \delta(g_i, g_j).$$

(3)

## Modularity maximization

In this study, we determine the optimal group assignment $g$ using the iterated version of the multiplex general Louvain algorithm [42] with default settings aside from 'moverandw', where a node moves to a new community when the probability of choosing that move is proportional to the increase in modularity that it results in. This setting helps to mitigate some of the undesirable behaviour in ordinal multiplex networks discussed in [43]. The iterated version of the general Louvain algorithm was used to reduce some of the inherent variability in individual runs present in modularity maximization [43,44], and is similar to consensus clustering [45]. Additionally, we further improve the stability of $Q$ by performing multiple runs as is standard in the literature [46–48]. Furthermore, the networks in this study are both small and weighted, where issues in $Q$ variability are much less severe [44]. It is of note that, while the Louvain method has been defined for both positive and signed networks [24], we choose to consider negative weights as equal to positive weights for modularity maximization. Negative weights have been argued to be neurobiologically relevant [24], but interpreting the differences between modules constructed from negative and positive weights separately is not well understood. We worried that per subject differences in the proportion of positive to negative weights could lead to large changes in modularity that would be difficult to interpret, especially for individual nodes. It is of note that, in all following mentions of maximised modularity, that we use the aforementioned iterated version of the algorithm with 100 repetitions, choosing the community assignment that corresponds with the highest modularity from those runs. For our data, we found modularity to be tightly distributed over each run.

## Experimental setup

In this section, we outline the methodology used for studying nodal modularity, and how this measure interacts with the stages of disease in our functional and structural networks.

**Modular behaviour of the VSTMBT.** Initially, we verify the behaviour of modularity for the shape and binding tasks of the VSTMBT. To accomplish this, we compare the modularity ($Q$) of our shape and binding networks to random and Stochastic Block Model (SBM) surrogate networks. The aim of this experiment is to verify that the VSTMBT exhibits community structure better than random and that $Q$ is similar or higher than a structured surrogate network (the SBM).

For this, we first construct our random networks by taking an edge in our network and randomly rewiring the edge using the Brain Connectivity Toolbox (BCT) [38]. Single-layer $Q$ is then maximized on the randomized shape and binding networks. We also considered the SBM, a variation of the random graph above but with defined community structure. In essence, it takes a defined set of community labels, and a symmetric matrix defining the probability of connections existing between nodes and randomly constructs a network under these conditions [49]. Here, we construct the SBM with freely available code [50] from our shape and binding networks using the group assignment acquired from maximising $Q$ on the non-surrogate networks. In this way we obtain structured, binary, surrogate networks from our shape and binding tasks. We then compare the modularity of our shape and binding networks against their surrogate counterparts to explore their modular behaviour.

**Comparison of *nQ* with other nodal graph measures.** To explore *nQ* as a measure, we compare it to several graph measures within three datasets. Specifically, we compare *nQ* against other measures of nodal community structure: single and multiplex versions of degree, clustering coefficient, and PageRank algorithms. Single-layer versions of these measures were calculated with the BCT. Multiplex versions of these measures are defined as follows. We calculate multiplex degree as standard, but with the addition of inter-layer edges. For the multiplex clustering coefficient, multiplex triangles are described across two layers, with 2 edges of the triangle existing in one layer and the remaining edge existing on the other (defined as a two-triangle). More formally, it is a measure of the ratio of the number of two-triangles and the number of one-triads (3 connected nodes on one layer with 2 edges) for a node. Refer to Eq (22) in [16] for further information. For multiplex PageRank, the Pagerank centrality of a node on one layer influences the centrality on another. In brief, we choose to use the combined multiplex PageRank algorithm defined in [51], where centrality in one layer adds bias to both strategies 1 and 2 in calculating the PageRank centrality of a node in a separate layer.

For each of our tasks, binding and shape (for our controls), we construct a dual-layer network from the encmaint and probe phases. We maximize multiplex modularity using the previously mentioned iterated method and, using this community assignment, we calculate multiplex *nQ* at each node using Eq (1). We then compare *nQ* to the other multiplex nodal measures of community structure, discussed above, and plot the per node scatter plots.

Next, we explore these comparisons for the two other publicly available datasets – the NKI-Rockland cohort and Zachary's Karate Club. See S2 Appendix for more detail. For the neuroimaging data (NKI-RS), we construct multiplex, dual-layer, fMRI-DTI networks for each subject by connecting nodes in the fMRI network to their spatial replicas in the DTI network. We maximize $Q_{multiplex}$ and calculate *nQ* in the same way as discussed in the prior paragraph for our dataset, and compare *nQ* to the same graph measures.

For Zachary's Karate Club, we maximize the single-layer version of *Q* using the iterated version in [42]. We then calculate single-layer *nQ* (Eq (3)) from the community assignment corresponding to the maximized *Q* in the previous step. Lastly, we compare the single-layer versions of degree, clustering coefficient, and PageRank to *nQ*.

**Application of *nQ* to investigate local-scale changes in MCI.** Here, we detail how comparisons are made between control and disease groups, single and multiplex network constructions, and binding and shape tasks to validate the utility of *nQ* in the characterisation of MCI.

For each of our subject groups (control, early MCI, MCI, and MCI converters), and for each of our shape (shape only) and binding tasks (coloured shapes), we explore single-layer encmaint and probe networks and their corresponding dual-layer network. Furthermore, we construct single-layer DTI networks for each subject.

For all of the above networks, we calculate *nQ* with the appropriate single or multiplex methods as described previously. We then compare the differences in *nQ* between control and disease for each network and task using two statistical tests: permutation test and receiver operating characteristic (ROC).

The permutation test is a method of hypothesis testing widely applicable to a variety of use cases [52]. In brief, given two sets of labelled data (such as control and disease), the permutation test computes the test statistic (in our case a two-sided test of the difference of means between the two samples) across many permutations of the labels of the two groups. For each permutation of the labels, we calculate the difference of means between the newly permuted groups and the difference of means for our original group. In total, our p-value is the proportion of sampled permutations where the absolute difference is different from the absolute

value of our original observed difference in means. In our study, we calculate p-values with 10000 permutations using code from [53].

The receiver operating characteristic (ROC) is a diagnostic measure of accuracy [54]. ROC plots are a measure of sensitivity vs. specificity, and typically, values of the area under the curve (*AUC*) generated this way is a measure between 0.5 (no apparent distributional difference between the two groups of test values) and 1 (perfect separation of the test values of the two groups).

We use the permutation test and ROC, for each of the models, to explore the effectiveness of *nQ* at categorizing the stages of disease in single vs. multiplex constructions and binding vs. shape tasks. We do this for each node in our network, comparing the values of *nQ* for healthy controls to each of our groups (early MCI, MCI, and MCI converters). That is, for each of the previously mentioned networks, we calculate permutation tests in the following way. For each node (brain region), we take the *nQ* calculated for each subject in the control group, compare the difference in means with the permutation test and the accuracy of the ROC in classifying disease and control groups to obtain the p-value (*p*), effect size, and area under the curve (*AUC*) of the ROC. Due to our small sample size, we take a conservative approach, reporting the regions where $p \leq 0.05$ and where p-values passed Benjamini-Hochberg false discovery rate (FDR) correction as regions exhibiting significant difference in *nQ* when comparing control and disease groups. FDR was conducted with the Multiple Testing Toolbox in Matlab [55] and visualizations of ROIs which passed this test and their statistical results are given as tables or visualized on a brain mesh using the BrainNet viewer [56].

FDR was used to control for the existence of potential false positives given our large number of statistical comparisons (85 in single-layer and 170 in dual-layer) [57], and plots of this correction for our results are given in S1 Fig. This was applied to our p-values at a threshold of $\alpha = 0.2$, indicating that we accept up to 20% of our statistically significant results to be false positives. We note that a requirement for FDR to properly control false positives is independent or positively dependent comparisons. Given that *nQ* is a nodal measures of group structure (within module comparisons are expected to be positively correlated) and *Q* increases across the stages of AD, we expect this requirement to be met in the majority of cases. However, due to the complexity of biological networks and AD's effect on brain network reorganization, we cannot rule out the possibility of some negative dependencies between p-values and acknowledge this as a limitation of this study. Alternative methods to control false positives for multiple comparisons under arbitrary dependency are either inappropriate for this study [58], or are known to be too conservative [59]. This is especially the case given that our small sample size limits the size of our p-values, limiting the effectiveness of applying conservative FDR approaches. However, [60] argues that discriminative power is more linked to effect size and variability than group size, motivating us to report the ROC AUC for completeness.

We perform a similar, per node analysis of our DTI data. Specifically, we explore *nQ* calculated on our single-layer FA-weighted DTI networks. We do the same group comparisons of control vs. disease with permutation test and ROC as in the previous section, reporting regions where $p \leq 0.05$ and where p-values survived FDR correction at $\alpha = 0.2$.

Lastly, we explore ROI changes in *nQ* for early MCI vs. MCI and MCI vs. MCI converters for our fMRI multiplex and DTI single-layer networks. This is done in the same manner as above by calculating the *nQ* for each of the ROIs in our networks, then comparing the differences in this measure with our statistical tests.

## Results

In this section, and following the methodologies described previously, we verify the behaviour of $Q$ for our task-fMRI, perform $nQ$ benchmarking, and apply this measure to our shape and binding task-fMRI and DTI to explore changes in $nQ$ along the AD continuum.

### Modular behaviour of the VSTMBT

Initially, we verify that $Q$ behaves as expected for our shape and binding tasks. We compare $Q$ in our shape and binding networks separately, and in random and SBM surrogate networks in Fig 3. Each of the shape and binding networks exhibited a better modular structure than random and similar modularity to our SBM surrogate networks, as expected. This verifies that both the shape and binding tasks of the VSTMBT exhibit a modular structure. Of note is the slightly increased modularity in the SBM surrogate networks. This is likely due to the binary weighting of the surrogate networks leading to a more defined group structure. Regardless, the modularity of the fMRI and SBM surrogate networks, along with the significantly improved performance over the random networks, suggests that the functional networks constructed from the shape and binding tasks exhibited modular group structure as expected.

### Independence of $nQ$ with other nodal graph measures

Next, we explore whether $nQ$ captures distinct information compared to other nodal graph measures for our dual-layer fMRI data, for each of our tasks, and in two additional datasets, NKI-RS and Zachary's Karate Club.

For our dual-layer fMRI networks and our cognitively normal subjects, we find that, for the binding task, there are moderate negative correlations between $nQ$ and degree and PageRank (–0.54 and –0.6, respectively), and a slight negative correlation with clustering coefficient as seen in Fig 4. Since fMRI networks are fully connected, we expect some high degree nodes to be too densely connected brain wide, rather than specialized to a specific functional grouping (resulting in lower $nQ$), which could explain the moderate negative correlation between degree and $nQ$.

In Zachary's Karate club, we find that single-layer $nQ$ is strongly correlated with Degree and PageRank ($r = 0.77$ and $r = 0.78$ respectively), but not with Clustering Coefficient

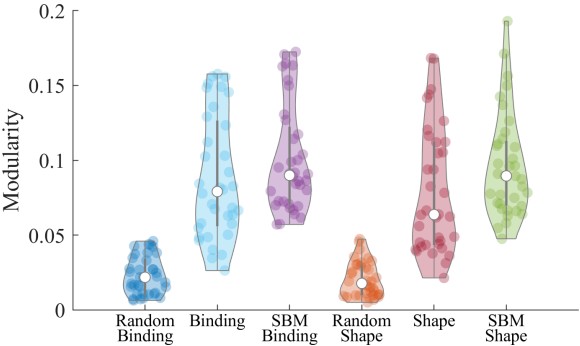

**Fig 3. Random and SBM null models for comparisons of modularity in shape and binding tasks.** Above are the violin plots of the modularity distribution for the shape and binding tasks. This is also explored for our random and SBM surrogate networks. Violin plots were generated with publicly available code [61].

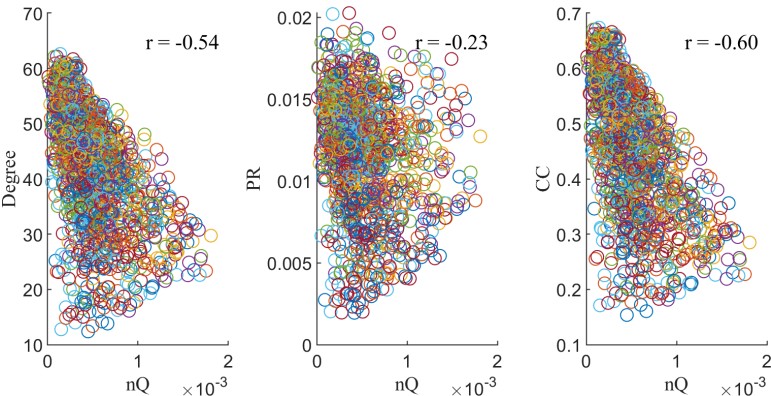

**Fig 4. *nQ* vs. other graph measures.** Scatter plots of Degree, PageRank (PR), and Clustering Coefficient (CC) vs. *nQ*. These measures were calculated in our dual-layer binding task-fMRI networks for cognitively normal subjects. The Pearson correlation between these comparisons is given by r.

($r$ = –0.27) as seen in Fig 5a. Since community structure in this network largely centres on the leaders of the two clubs, this could explain the similarity between the degree of a node and its *nQ* (contribution to group structure), while PageRank behaving similar to degree in this case could be a result of the network being fairly regular (each node in the network having a similar degree).

In NKI-RS, for the rs-fMRI and DTI dual-layer networks, we find a fairly strong correlation ($r$ = 0.69) between *nQ* and degree, while close to no correlation ($r$ = 0.07) in PageRank and a moderate negative correlation ($r$ = –0.53) in Clustering Coefficient as seen in Fig 5b. Interestingly, the relationship between *nQ* and clustering coefficient seems to highlight the difference in modularity between rs-fMRI and DTI. In this dataset, the modularity of the DTI layer is much higher than the fMRI layer. Therefore, we would expect that the nodes that

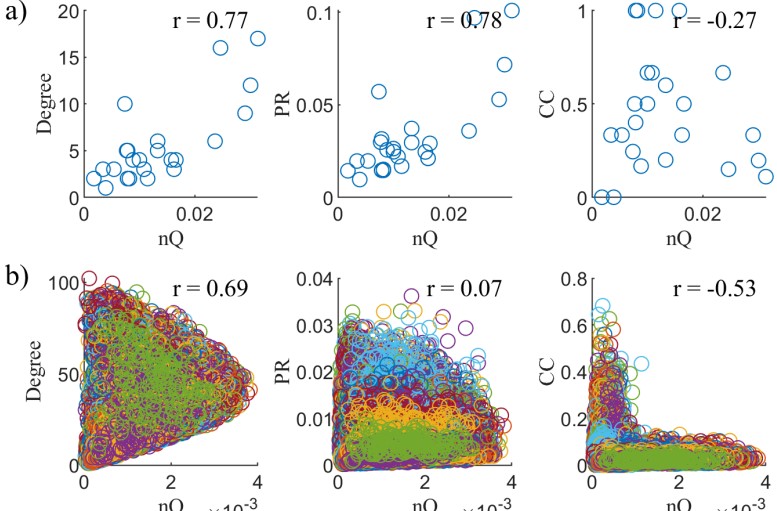

**Fig 5. Behaviour of *nQ* in ZKC and NKI.** Scatter plots of Degree, PageRank (PR), and Clustering Coefficient (CC) vs. *nQ* for Zachary's Karate club (a) and NKI-RS (b). r is the Pearson correlation coefficient.

interact most with the DTI layer rather than the fMRI layer would result in the highest $nQ$. Since multiplex triangles require integration between both layers, a high multiplex clustering requires strong integration with the fMRI layer which could result in a decreased $nQ$. This is in contrast to a strongly connected node in the DTI layer with a weak connection to the fMRI layer resulting in high $nQ$ and low multiplex clustering. This could explain the behaviour seen in CC vs. $nQ$ in Fig 5b.

In sum, all three datasets (with single and dual-layer constructions), exhibited different relationships between $nQ$ and the various graph measures. While not an exhaustive list, these tests, along with the formulation of $nQ$ such that $\sum nQ = Q$, add reassurance to the independent and novel behaviour of $nQ$ as a measure for exploring granular group structure in networks. Additionally, studies which calculate $nQ$ either in low complexity networks or multiplex networks where layers contain a high difference in structure and/or network density should consider the potential overlap between the measures of node influence explored here. In these instances, computationally more efficient measures may be beneficial to prioritize while remaining similarly informative.

## Application of *nQ* to explore local-scale changes in MCI

Here, we explore the changes in $nQ$ across the stages of AD using the previously described permutation test and the area under the curve of the ROC, reporting regions that pass the thresholds of $p \leq 0.05$ and where $p$ is FDR controlled at $\alpha = 0.2$.

**fMRI.** First we explore comparisons between controls and eMCI, MCI and MCI converters. We find that in single and multiplex constructions, for only the binding task and not shape, that the number of regions that exhibit statistically significant changes in $nQ$ occur only for our comparisons between control vs. MCI converters (25 ROIs across encmaint and probe task phases exhibited abnormal $nQ$ for multiplex, 20 ROIs for single-layer). See Fig 6c for a visualization of these ROIs for the multiplex model and see S1 Table for a comparison between single and multiplex models. For our multiplex binding model, we find moderately low p-values (**range: [0.001 - 0.029]**) and high effect sizes (**range: [1.149 - 2.363]**) and ROC AUC (**range: [0.792 - 0.979]**), see Table 2. Note that since we are applying FDR at $\alpha = 0.2$ that up to 20% of our ROIs may be false positives (equating to 5 ROIs in the multiplex model and 4 ROIs in the single-layer case). Furthermore, given that no ROIs survived FDR correction for the shape task (for both single and multiplex models) and multiplex modelling resulted in improved performance, we focus on reporting fMRI results for the binding task as a multiplex network. Additionally, in our comparisons of eMCI vs. MCI and MCI vs. MCI converters for fMRI shape and binding, no p-values survived FDR correction.

**DTI.** Next we look at changes in white matter microstructure in our DTI networks for comparisons between controls and our disease groups. We find, like in our fMRI networks, that the largest number of ROIs that exhibit statistically significant changes in $nQ$ occurs in our comparison of controls and MCI converters as expected (4 ROIs for controls vs. MCI converters and 2 ROIs for controls vs. MCI), shown visually in Fig 7. These ROIs also exhibited moderately low p-values (**range [0.001 - 0.007]**) and high effect sizes (**range [1.504 - 2.390]**) and ROC AUC (**range [0.863 - 0.958]**), as seen in Table 3.

Unlike our comparisons between early MCI vs. MCI for our fMRI models, we find that two ROIs pass our statistical thresholds for our DTI networks. Specifically, the right postcentral and precentral which were also observed in our prior comparisons of control vs. disease, see Fig 8a. As before, these ROIs exhibited moderately low p-values and high effect sizes and ROC AUC, see S2 Table.

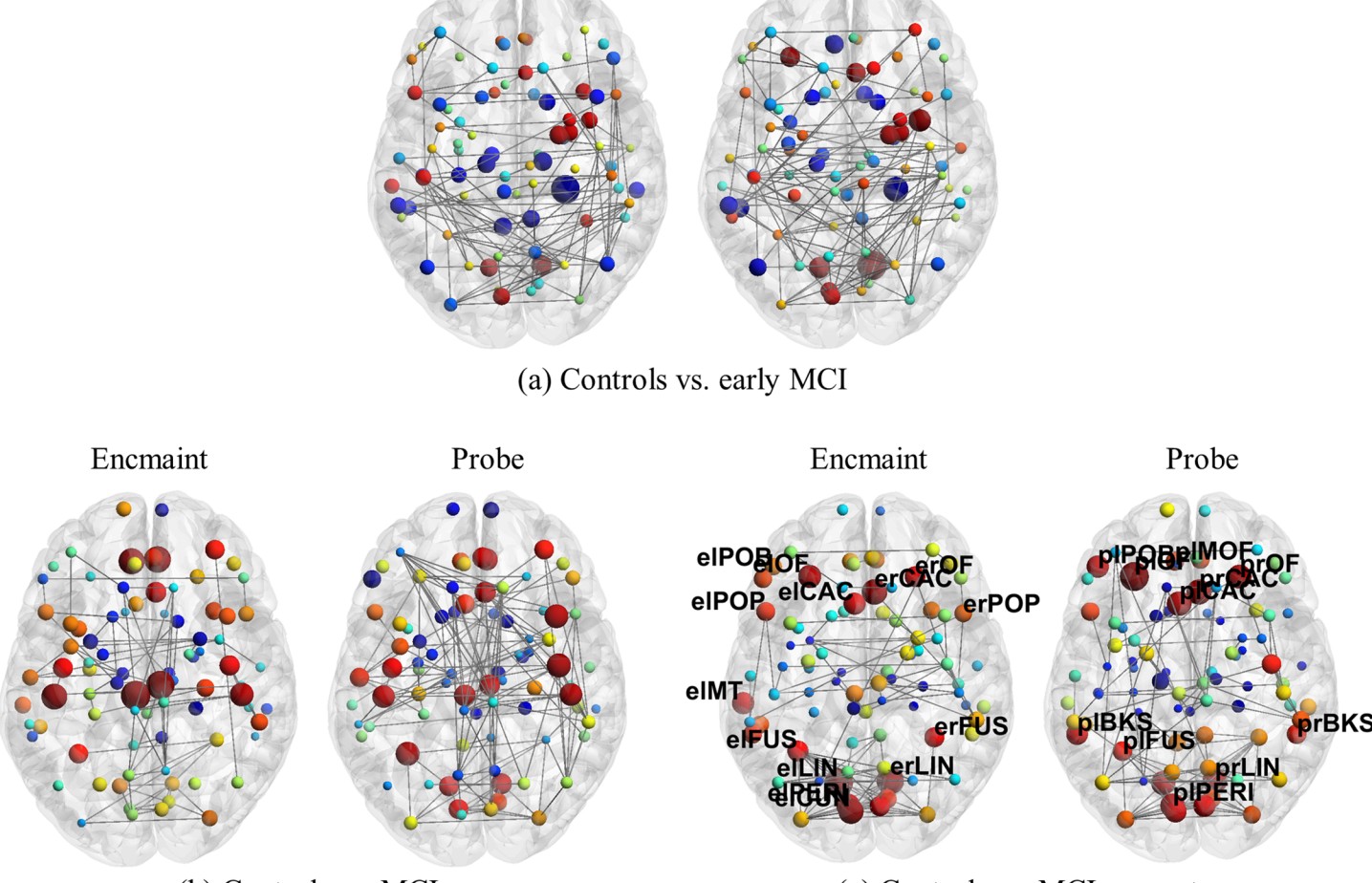

**Fig 6. Changes in *nQ* for multiplex fMRI binding.** Using BrainNet Viewer, we visualize encmaint and probe (left and right brain respectively) layers of our network. Here, nodes in blue represent loss of *nQ* while those in red represent gains in *nQ*. The size of the nodes represent the magnitude of this change, while labelled nodes are those that passed $p \leq 0.05$ and FDR controlled at $\alpha = 0.2$. Labels follow the form task phase (e or p), followed by brain hemisphere (l or r), then a shortened version of the ROI (i.e. LIN refers to the lingual). Refer to Table 2 for a more detailed breakdown of the ROIs present in this figure. Furthermore, only 1.5% of edges are visualized for clarity. Note the magnitude of the gains in *nQ* for the later stage disease comparisons.

# Discussion

The following sections explore how *nQ* reflects aspects of neuropsychology, multiplexity, and neuroanatomy in AD.

## Binding and Shape tasks

Parra et al. [7] found that when the demands of the tasks are adjusted to the capacity of the patients, the binding condition was highly sensitive to the disease, while the performance on the shape-only task did not change significantly between controls and familial AD. Our results strongly confirm this behaviour. We were unable to detect any statistically significant results for our shape task across all control and disease comparisons, while observing a high number of ROIs exhibiting abnormal *nQ* in the binding task for both the single-layer and

Table 2. fMRI multiplex binding for controls vs. MCI converters. This table displays the ROIs which passed $p \leq 0.05$ and where p-values are controlled by FDR at $\alpha = 0.2$. These ROIs reside in either the encmaint (EM) or probe (P) layers of our multiplex network and in left (L) or right (R) hemispheres of the brain. Standard p-value and effect size is displayed following permutation test and the area under the curve (AUC) of the Receiver Operating Characteristic (ROC).

| ROIs | p | effect size | ROC AUC |
|---|---|---|---|
| EM R-caudalanteriorcingulate | 0.002 | 2.362 | 0.938 |
| EM L-caudalanteriorcingulate | 0.006 | 1.924 | 0.875 |
| EM L-lingual | 0.007 | 1.883 | 0.896 |
| EM R-lingual | 0.010 | 1.774 | 0.896 |
| EM L-middletemporal | 0.008 | 1.708 | 0.896 |
| EM L-pericalcarine | 0.012 | 1.659 | 0.854 |
| EM R-parsopercularis | 0.007 | 1.634 | 0.813 |
| EM R-lateralorbitofrontal | 0.006 | 1.624 | 0.875 |
| EM L-parsorbitalis | 0.010 | 1.614 | 0.875 |
| EM L-cuneus | 0.014 | 1.556 | 0.896 |
| EM L-parsopercularis | 0.016 | 1.483 | 0.833 |
| EM L-fusiform | 0.020 | 1.453 | 0.792 |
| EM R-fusiform | 0.029 | 1.394 | 0.813 |
| EM L-lateralorbitofrontal | 0.026 | 1.149 | 0.792 |
| P R-caudalanteriorcingulate | 0.001 | 2.363 | 0.979 |
| P L-caudalanteriorcingulate | 0.003 | 2.039 | 0.917 |
| P L-bankssts | 0.001 | 1.722 | 0.958 |
| P L-lateralorbitofrontal | 0.003 | 1.702 | 0.917 |
| P R-lateralorbitofrontal | 0.006 | 1.663 | 0.896 |
| P L-parsorbitalis | 0.002 | 1.64 | 0.938 |
| P L-pericalcarine | 0.012 | 1.626 | 0.833 |
| P R-lingual | 0.015 | 1.531 | 0.854 |
| P L-fusiform | 0.014 | 1.453 | 0.854 |
| P L-medialorbitofrontal | 0.026 | 1.444 | 0.813 |
| P R-bankssts | 0.027 | 1.382 | 0.833 |

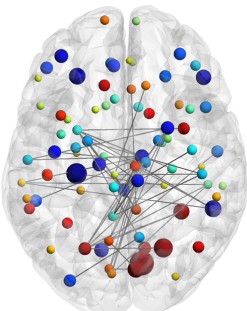
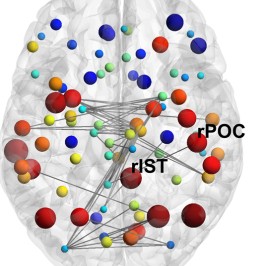
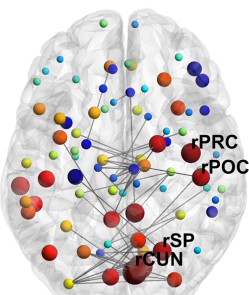

(a) Controls vs. early MCI　　　　(b) Controls vs. MCI　　　　(c) Controls vs. MCI converters

**Fig 7. Changes in *nQ* for single-layer DTI.** As before, blue indicates a loss of *nQ* while red represents a gain. Here we also see an increase (node size) in *nQ* for later stage disease comparisons. Furthermore, 1.5% of the network edges are displayed for clarity. Additionally, to reiterate, the DTI networks are constructed based on white matter microstructure. As such, we have a static (no temporal element like in the fMRI networks) single-layer network in this case where the single layer of the network is represented visually by one brain.

multiplex models when comparing controls vs. MCI converters. These results indicate that *nQ* may be detecting binding task-specific changes in our brain networks at a crucial turning point of AD.

**Table 3. DTI for control vs. disease.** This table displays the ROIs which passed the thresholds of $p \leq 0.05$ and where the p-values were FDR controlled at $\alpha=0.2$. L and R indicate the left or right hemispheres of the brain respectively. Standard p-value and effect size is displayed following permutation test and the area under the curve (AUC) of the Receiver Operating Characteristic (ROC).

| Controls vs. | ROIs | p | effect size | ROC AUC |
|---|---|---|---|---|
| MCI | R-postcentral | 0.004 | 1.637 | 0.863 |
| | R-isthmuscingulate | 0.003 | 1.539 | 0.888 |
| MCI converters | R-cuneus | 0.001 | 2.390 | 0.958 |
| | R-precentral | 0.001 | 2.046 | 0.958 |
| | R-superiorparietal | 0.002 | 1.730 | 0.938 |
| | R-postcentral | 0.007 | 1.504 | 0.875 |

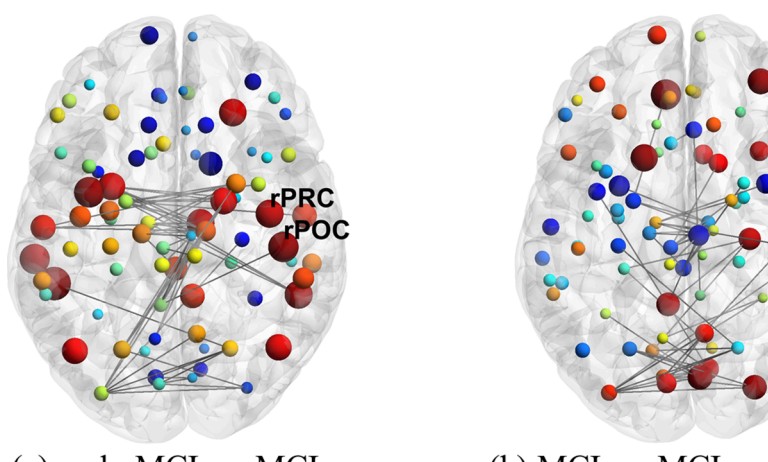

(a) early MCI vs. MCI          (b) MCI vs. MCI converters

**Fig 8. Comparisons of *nQ* for early MCI vs. MCI and MCI vs. MCI converters for single-layer DTI.** Figure generation and details follow from Fig 7. a) Here note the large (node size) increase (red nodes) in *nQ* in the parietal lobe.

## Single-layer vs. multiplex networks

We explored whether modelling the two task phases (encmaint and probe) as a multiplex temporal network yields a greater sensitivity to AD. We found that multiplex modelling resulted in a higher number of ROIs which pass our statistical thresholds (25 vs. 20 for single-layer), and showed generally improved p-values, effect sizes, and ROC AUC. This is in agreement with other multiplex temporal network studies which show that multiplex constructions yield insights beyond the analysis of each layer individually [17,18,37]. We expect that network models of AD using *nQ* that were constructed with additional layers would see even more drastic differences when comparing single-layer and multiplex network constructions. However, the benefits of this approach and its connection to diseases such as AD require further study.

## Neuroanatomical exploration of fMRI

Before exploring the neuroanatomical implications of our results, we first explore the interpretation of *nQ* in this setting. When considering *nQ* in the context of graph theory, an increase in *nQ* can indicate an increased involvement or specialization of a node within its

own group and/or a reduction in between group connectivity. For the brain, this can be interpreted as a combination of reduced communication between brain subnetworks, and/or increased activity within key brain regions. Similarly, an increase in $nQ$ in our DTI networks would reflect structural reorganization. Either an increase in white-matter microstructure between a specific ROI and those within its structural group, or a reduction in between group connectivity. When comparing healthy and impaired brain networks, these changes in $nQ$ convey nodal aspects of functional and structural reorganization due to damage or disease. Such compensatory functional reorganization has been reported in the early stages of AD for both MRI [62] and EEG [63]. Additionally, many key sub-networks, like the default mode network, have shown to be key in understanding the brain at rest [64]. $nQ$ could prove to be invaluable in understanding the role of individual ROIs both within and between their sub-networks.

Previously, we hypothesized that since $Q$ has been shown to increase over the stages of AD in fMRI [26], that a more granular analysis of AD with $nQ$ would uncover key insights into the individual stages of AD. Additionally, the binding task targets visual memory binding and the temporary retention of complex objects (coloured shapes in this case) and we hypothesized that key ROIs integral to these processes would be impacted, given that the VSTMBT is a cognitive biomarker of AD. Our results reflect this (see Fig 6 and Table 2).

We observe extensive changes when comparing controls to MCI converters and explore the fine grain changes in $nQ$. Specifically, we find dysfunction centred in the limbic, paralimbic, and visual systems - lingual, caudal anterior cingulate, lateral orbitofrontal, fusiform, cuneus, pericalcarine, and medial orbitofrontal. These functional changes in $nQ$ in the visual, limbic, and paralimbic systems comprise 18/25 (72%) of the ROIs identified (note that we focus on general changes given that up to 5 ROIs may be false positives attributed to FDR correction at $\alpha$ = 0.2). Our findings suggest a potential increased activity in visual, limbic, and paralimbic systems and/or reduced communication between these subnetworks and others within the brain. These increases in $nQ$ may be driven by impairments in brain wide communication between or within functional groups as compensation for performance deficits due to AD. Notably, the increases in $nQ$ in the visual, limbic, and paralimbic systems are also common locations for the accumulation of tau and neurofibrillary tangles associated with Braak stages III-IV where MCI typically becomes observable [65]. Additionally, recent research in the VSTMBT suggests that the lingual, fusiform, middle temporal, and pericalcarine are areas of increased amyloid-$\beta$ deposition in cognitively unimpaired adults with poor memory binding [66], creating a compelling connection between the VSTMBT and a common biomarker of AD. This is especially interesting given that these regions were all seen in the encmaint phase (where binding occurs), but not all in probe (missing the middle temporal, and cuneus) as expected. This connection between amyloid-$\beta$ deposition and poor memory binding performance may explain some of the increased compensatory functional reorganization ($\uparrow nQ$) in these regions.

It should be noted that, while Braak staging is well described at a population level [67,68], recent research suggests that there is variation at the individual level and that this trajectory appears to vary along at least four archetypes of tau spread in the brain [67]. We acknowledge this as a limitation and encourage the use of $nQ$ in exploring AD sub-typing and connection to tau deposition in larger, multi-modal, datasets. Despite this, the connection between observed ROIs with classical tau deposition and amyloid-$\beta$ remains encouraging. Additionally, it is important to highlight the stark difference when comparing the results from our comparisons of controls vs. MCI and controls vs. MCI converters (0 vs. 25 ROIs). The differentiation between subjects with MCI and those who will convert to AD (in this case after 2 years) is important both for understanding the disease and in its early detection before

damage has been done. While limited by sample size, this stark contrast in our comparisons between controls vs. MCI and MCI converters warrant further exploration of $nQ$ in larger datasets.

### Neuroanatomic exploration of DTI

Similar to our fMRI results, we find that the number of ROIs that exhibited changes in $nQ$ increased with disease severity (2 in control vs. MCI and 4 in control vs. MCI converters), including increases in effect sizes and decreases in p-values. DTI results were fairly distinct from fMRI aside from the cuneus which was identified when comparing controls vs. MCI converters in both modalities (however note the difference in laterality - left cuneus in fMRI and right cuneus in DTI). Interestingly, it is specifically the right cuneus that has been observed to have increased amyloid-$\beta$ deposition among adults with poor memory binding, complimenting our fMRI results [66]. Additionally, [69] found that Braak staging was associated with reduced FA in many of the regions, and specifically in limbic pathways connecting the medial temporal lobe to subcortical grey matter and medial parietal lobes. Reduction in FA in pathways connecting the medial temporal lobe to the parietal lobe could directly result in increased $nQ$ in these regions, which we observed in the parietal lobe (post central and superior parietal). Such changes have been found in the brains of patients with MCI who are at the highest risk to progress to dementia. For example, Parra et al. [10], observed that white matter integrity of the frontal lobe in carriers of the mutation E280A of the PSEN-1 gene [70] correlated with performance when they were assessed with either an associative memory task (i.e., a form of relational binding in long-term memory, see [7] for details) or the VSTMBT. While we observed an increase in $nQ$ in one frontal region (precentral gyrus), perhaps the most interesting finding is the laterality across our comparisons (all identified ROIs were in the right hemisphere). Parra et al. [10] observed that the genu of the corpus callosum accounted for VSTMB impairments and the hippocampal part of the cingulum bundle accounted for long-term memory binding deficits. The corpus callosum is often cited as an interface for cross-hemispheric communication in the brain, and damage to this structure could result in right dominant structural reorganization and reflected in our $nQ$ results.

### Conclusions

In neuroimaging, modularity has been largely limited by its use as a global metric. This is despite its utility in describing biological networks such as the brain, for which modularity is a core characteristic. Changes in modularity can be observed as the brain responds to task stimuli by dynamically reorganizing itself over time and when compensating for diseases such as AD [24,26]. However, this reorganization is not yet fully understood.

   To tackle this, we introduced $nQ$ as a method to measure the specific contribution of an ROI to modularity. We explored this measure in the VSTMBT, which is known to require functional activity in specialized and distinct brain regions, along with communication between these regions, to drive cognitive functions such as encoding and retrieval in short-term memory binding [27]. We found that $nQ$ captured the fine grain changes in visual, limbic, and paralimbic functional networks, and were in agreement with tau and amyloid-$\beta$ deposition for poor memory binders. This trajectory was further supported in our DTI networks where results complimented previously understood changes in white matter integrity in poor memory binders in frontal and parietal lobes. Furthermore, $nQ$ was able to distinctly differentiate two key stages of AD (MCI from MCI converters), encouraging further study of $nQ$ as a potential diagnostic measure of AD. While limited by a small sample size, the results were

consistent across hypotheses relating to single and multiplex constructions, DTI and fMRI, AD biomarker locations, and shape and binding tasks.

As changes in modularity are also observed in EEG and MEG [71,72], future analyses of these granular changes using $nQ$ could not only lead to a greater understanding of how functional reorganization occurs in AD, but have the potential in characterizing disease stages in a more widely available imaging modality (EEG). Additionally, given the wide spread use of classical modularity in the research of real world networks, we expect $nQ$ to improve these analyses by providing insight into local-scale group structure. For instance, in the exploration of granular group structure in protein-protein interaction, ecosystems, and many other biological networks [73], in understanding how individuals in static and dynamic interaction networks influence disease spread [74], or in improving the resilience of power grids by identifying vulnerable nodes to mitigate the impact of cascade failures [75]. Lastly, it is of interest to study how $nQ$ behaves when calculated from group assignments that are not derived from modularity maximization. $nQ$ may be able to be used in conjunction with community detection algorithms that bring improvements in group segregation or computational efficiency, expanding $nQ$'s utility to general community detection domains by providing an improved measure of node influence.

In sum, the results of this research motivate further study of $nQ$ in AD, network-based analyses of the VSTMBT, and the application of $nQ$ to other diseases and networks beyond those biological to understand changes in modularity at local and mesoscales.

## Supporting information

**S1 Appendix. fMRI windowing and the variable maintenance phase of the VSTMBT.**
(PDF)

**S2 Appendix. Description of publicly available datasets.**
(PDF)

**S1 Fig. FDR plots.** This figure visualizes the Benjamini-Hochberg FDR correction for each of our comparisons. Plots of observed p-values ($p_k$) by rank ($k$) show the ranked p-values in our ROI comparisons of a) fMRI multiplex control vs. MCI converters, b) DTI control vs. MCI, c) DTI control vs. MCI converters, and d) DTI eMCI vs. MCI. FDR controlled p-values are all $p_i$ from $i = 1...k$ where $p_k \leq \frac{\alpha k}{m}$ and $\alpha$ is our chosen threshold. Plots of FDR adjusted p-values are also given to more easily visualize those that pass FDR correction (those that fall under the dashed line at $\alpha = 0.2$).
(TIF)

**S1 Table. Single-layer vs. multiplex results for the binding task and control vs. MCI converters.** This table displays the ROIs which pass the thresholds of $p \leq 0.05$ and where p-values are controlled by FDR at $\alpha = 0.2$. These ROIs reside in either the encmaint (EM) or probe (P) single-layers or modeled together as a multiplex network (results given seperately in the left and right sides of the table respectively). Standard p-value (p) and effect size is displayed following permutation test and the area under the curve (AUC) of the Receiver Operating Characteristic (ROC). Multiplex construction yields a higher number of ROIs which pass the statistical thresholds (25 vs. 20), with lower p-values and higher effect sizes and ROC AUC among most ROIs. Additionally, note that FDR is applied to each layer separately in the single-layer case (85 ROIs each). Consequently, and since FDR is more strict for a higher number of comparisons, the multiplex model displays higher statistical power in determining regional changes in $nQ$ between controls and MCI converters.
(TIF)

**S2 Table. Changes in *nQ* in DTI for early MCI vs. MCI and MCI vs. MCI converters.** This table displays the ROIs which passed $p \leq 0.05$ and FDR controlled at $\alpha = 0.2$. L and R indicate the left or right hemispheres of the brain respectively. Standard p-value and effect size is displayed following permutation test and the area under the curve (AUC) of the Receiver Operating Characteristic (ROC).
(TIF)

**S3 Table. Neurospsychological tests.** S3 Table shows the neuropsychological profile of MCI patients, early MCI patients and healthy controls entering the study. ANOVA revealed that patients with MCI performed poorer than healthy controls on ACE [76], MMSE [76], HVLT-DELAY and TOT [77], FAS [78], DIGIT SYMBOL [79], REY-IMMEDIATE and DELAY [80], GNT [81], CLOCK [76], FCSRT-IFR and ITR [82], and TOPF [83]. More specifically, significant differences emerged from comparisons between MCI patients and healthy controls, and MCI patients versus early MCI patients overall. HVLT was carried out poorly from both MCI and early MCI patients compared to the control group, whereas Rey figure delayed copy was significantly underperformed by MCI patients only.

Although the conversion to AD in some patients has been ascertained once the collection of neuropsychological data was done, and MCI converters have not been taken into account here, we can conclude that these results are in line with clinical diagnosis and reflect the progression of the disease through the spectrum.
(TIF)

## Acknowledgments

We are thankful for the support of NHS Scotland (both Lothian and Forth Valley boards) in recruiting MCI patients and the Volunteer Panel for healthy controls at the University of Edinburgh. Additionally, we thank the reviewers for their feedback which significantly improved this manuscript.

## Author contributions

**Conceptualization:** Avalon Campbell-Cousins, Javier Escudero.

**Data curation:** Avalon Campbell-Cousins, Federica Guazzo, Mark E. Bastin, Mario A. Parra.

**Formal analysis:** Avalon Campbell-Cousins, Federica Guazzo, Mario A. Parra.

**Funding acquisition:** Mario A. Parra, Javier Escudero.

**Investigation:** Avalon Campbell-Cousins.

**Methodology:** Avalon Campbell-Cousins.

**Resources:** Mark E. Bastin, Mario A. Parra.

**Software:** Avalon Campbell-Cousins.

**Supervision:** Javier Escudero.

**Validation:** Avalon Campbell-Cousins.

**Visualization:** Avalon Campbell-Cousins.

**Writing – original draft:** Avalon Campbell-Cousins.

**Writing – review & editing:** Avalon Campbell-Cousins, Federica Guazzo, Mark E. Bastin, Mario A. Parra, Javier Escudero.

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
