## [Decision Letter · Decision Letter 0]

21 Apr 2025

PONE-D-25-03889Multiplex Nodal Modularity: A novel network metric for the regional analysis of amnestic mild cognitive impairment during a working memory binding taskPLOS ONE

Dear Dr. Campbell-Cousins, Thank you for submitting your manuscript to PLOS ONE. After careful consideration, we feel that it has merit but does not fully meet PLOS ONE’s publication criteria as it currently stands. Therefore, we invite you to submit a revised version of the manuscript that addresses the points raised during the review process.

We look forward to receiving your revised manuscript.

Kind regards,

Stephen D. Ginsberg, Ph.D.

Section Editor

PLOS ONE

**Journal requirements:** 1. When submitting your revision, we need you to address these additional requirements. Please ensure that your manuscript meets PLOS ONE's style requirements, including those for file naming. The PLOS ONE style templates can be found at https://journals.plos.org/plosone/s/file?id=wjVg/PLOSOne_formatting_sample_main_body.pdf and https://journals.plos.org/plosone/s/file?id=ba62/PLOSOne_formatting_sample_title_authors_affiliations.pdf 2. Thank you for stating the following financial disclosure: ACC is funded by the Principal’s Career Development Scholarship (PCDS) for his PHD from the University of Edinburgh. This funder spent no role in the manuscript's creation/data collection. https://institute-academic-development.ed.ac.uk/postgraduate/doctoral/career-management/principals-scholarshipsMAP received funding from the Alzheimer’s Society towards the Longitudinal Study of MCI through the Grants AS-R42303 and AS-SF-14-008. This funding was for the collection of MCI data (used in the manuscript) and specifically in conjunction with this research article 10.1186/s13195-022-01082-9. For the previously mentioned article, the funders only supported the study (proposed research) financially and monitored progress.https://www.alzheimers.org.uk/   Please state what role the funders took in the study.  If the funders had no role, please state: "The funders had no role in study design, data collection and analysis, decision to publish, or preparation of the manuscript." If this statement is not correct you must amend it as needed. Please include this amended Role of Funder statement in your cover letter; we will change the online submission form on your behalf. 3. Thank you for stating the following in the Acknowledgments Section of your manuscript: Avalon Campbell-Cousins was supported by Edinburgh University’s Principle’s Career Development PhD Scholarship. The authors also acknowledge the support from the Alzheimer’s Society towards the Longitudinal Study of MCI through the Grants AS-R42303 and AS-SF-14-008 awarded to MAP. Additionally, the support of NHS Scotland (both Lothian and Forth Valley boards) in recruiting MCI patients and the Volunteer Panel for healthy controls at the University of Edinburgh. We note that you have provided funding information that is not currently declared in your Funding Statement. However, funding information should not appear in the Acknowledgments section or other areas of your manuscript. We will only publish funding information present in the Funding Statement section of the online submission form. Please remove any funding-related text from the manuscript and let us know how you would like to update your Funding Statement. Currently, your Funding Statement reads as follows: ACC is funded by the Principal’s Career Development Scholarship (PCDS) for his PHD from the University of Edinburgh. This funder spent no role in the manuscript's creation/data collection. https://institute-academic-development.ed.ac.uk/postgraduate/doctoral/career-management/principals-scholarshipsMAP received funding from the Alzheimer’s Society towards the Longitudinal Study of MCI through the Grants AS-R42303 and AS-SF-14-008. This funding was for the collection of MCI data (used in the manuscript) and specifically in conjunction with this research article 10.1186/s13195-022-01082-9. For the previously mentioned article, the funders only supported the study (proposed research) financially and monitored progress.https://www.alzheimers.org.uk/  Please include your amended statements within your cover letter; we will change the online submission form on your behalf. 4. We note that you have indicated that there are restrictions to data sharing for this study. PLOS only allows data to be available upon request if there are legal or ethical restrictions on sharing data publicly. For more information on unacceptable data access restrictions, please see http://journals.plos.org/plosone/s/data-availability#loc-unacceptable-data-access-restrictions.  Before we proceed with your manuscript, please address the following prompts: a) If there are ethical or legal restrictions on sharing a de-identified data set, please explain them in detail (e.g., data contain potentially identifying or sensitive patient information, data are owned by a third-party organization, etc.) and who has imposed them (e.g., a Research Ethics Committee or Institutional Review Board, etc.). Please also provide contact information for a data access committee, ethics committee, or other institutional body to which data requests may be sent. b) If there are no restrictions, please upload the minimal anonymized data set necessary to replicate your study findings to a stable, public repository and provide us with the relevant URLs, DOIs, or accession numbers. For a list of recommended repositories, please seehttps://journals.plos.org/plosone/s/recommended-repositories. You also have the option of uploading the data as Supporting Information files, but we would recommend depositing data directly to a data repository if possible. We will update your Data Availability statement on your behalf to reflect the information you provide. 5. We notice that your supplementary tables are included in the manuscript file. Please remove them and upload them with the file type 'Supporting Information'. Please ensure that each Supporting Information file has a legend listed in the manuscript after the references list.

**Additional Editor Comments:**

After careful consideration by 2 Reviewers and an Academic Editor, all of the critiques of the Reviewers must be addressed in detail in a revision to determine publication status. If you are prepared to undertake the work required, I would be pleased to reconsider my decision, but revision of the original submission without directly addressing the critiques of the Reviewers does not guarantee acceptance for publication in PLOS ONE. If the authors do not feel that the queries can be addressed, please consider submitting to another publication medium. A revised submission will be sent out for re-review. The authors are urged to have the manuscript given a hard copyedit for syntax and grammar.

Reviewers' comments:

Reviewer's Responses to Questions

**Comments to the Author**

1. Is the manuscript technically sound, and do the data support the conclusions?

Reviewer #1: Partly

Reviewer #2: Yes

2. Has the statistical analysis been performed appropriately and rigorously?

Reviewer #1: Yes

Reviewer #2: Yes

3. Have the authors made all data underlying the findings in their manuscript fully available?

Reviewer #1: No

Reviewer #2: No

4. Is the manuscript presented in an intelligible fashion and written in standard English?

Reviewer #1: Yes

Reviewer #2: Yes

5. Review Comments to the Author

**Reviewer #1:** In this paper, Authors propose a new metric for assessing community structure in various single-and multi-layer networks, called nodal modularity (nQ). Contrary to the more traditionally used modularity, nQ not only quantifies community structure at a global level in a network, but also provides some information about community structure at a meso-scale level (i.e. at level of single regions). NQ was tested for calculation using functional MRI (fMRI) and diffusion tensor MRI (DT MRI) data from a group of mild cognitive impairment (MCI) patients and healthy controls, calculating multiplex nQ across functional/structural layers. Results suggest that observed changes in nQ in MCI align with expected trajectories of changes in alzheirmer’s disease, suggesting applicability of nQ as a biomarker for the disease.

The paper is of interest, but it overall very long, with several redundant and unnecessary information being reported; several sections can be considerably condensed. Following major issues should be addressed.

1) The Abstract repeatedly states that nQ is very useful (compared to standard modularity) because it is a more fine-grained metric giving information not only at global level but a mesoscale level. Despite this, the results reported in the Abstract are very generic and do not describe which are the regions showing abnormal nQ. Since Authors claim that this is one of the major novelty of their study, I think that results should be expanded in this direction.

2) The first section of the introduction is excessively long and includes a lot of information about Alzheimer's pathology (unrelated to neuroimaging) that is mostly background and not necessary for the paper. This section should be shortened. In general, the methods section is also overly detailed and can be significantly condensed. The same applies to the number of figures and tables.

3) The end of the Introduction is not really introduction, it anticipates already results and conclusions. All the last paragraph of the introduction reporting results and conclusions should be deleted.

4) One of the main limitations of the paper is the very small sample size of HC and MCI groups included. This should be clearly stated in the paper.

5) It is not clear which version of SPM software was used for fMRI analysis, some paragraphs mention SPM8 and other paragraphs mention SPM12. SPM8 is a very old software, so it is surprising that Authors use it for some analysis. To be sure that Authors are using appropriate software for the analysis, please also report software versions for Freesurfer and FSL.

6) The paragraph in the method about modularity is very long and can be condensed. The same applies for the description of the very standard network metrics (e.g. degree, clustering coefficient, centrality..). these are all very standard metrics which were described in details in hundreds of previous publications.

7) I don’t understand how the fact that nQ is not correlated with the other graph metrics constitutes a "verification" of nQ. It only tells us that it is independent, but it certainly does not ensure that it is meaningful.

8) Authors should provide a Results section clearly separated from the Discussion section. In the present paper version, the two sections are combined in a very confusing way.

9) Please delete the wording “disease progression” from any result and interpretation; “progression” implies a longitudinal assessment over time, while the present study is cross-sectional.

**Reviewer #2:** This study introduces nodal modularity (nQ), a novel network metric designed to evaluate particular changes in brain network community architecture, particularly in the context of amnestic mild cognitive impairment (aMCI) and Alzheimer's Disease (AD). The approach employed in this research is both captivating and technically solid, providing significant insights for the realms of neuroimaging and network neuroscience. The authors receive commendation for merging functional and structural data in an innovative analytical model and validating their techniques with established datasets.

However, I would like to highlight several concerns and limitations that I believe require the authors' attention and clarification before this manuscript.

Major Concerns

1. Limited Sample Size and Power

The number of subjects across groups — particularly in the early MCI and MCI converter subgroups — is relatively small. This naturally limits the statistical power and raises concerns about the robustness of the results. While the authors apply permutation testing and ROC analysis, the reliability of node-level inferences across so many comparisons is still questionable in the absence of adequate correction for multiple comparisons.

2. Absence of Multiple Comparison Correction

Given the large number of regions (85 ROIs) evaluated independently for significance, the study is at risk for inflated false positives. The authors have not clearly stated whether false discovery rate (FDR) correction or any other multiple testing correction was applied. For a study aiming to identify diagnostic biomarkers, statistical stringency is crucial.

3. Biological Interpretability of nQ

While nQ is mathematically well-defined, its biological interpretation remains somewhat abstract. It is not clear how variations in nQ values across brain regions translate into functional implications or pathophysiological mechanisms. A clearer, neurobiologically grounded discussion of what an increase or decrease in nQ implies would significantly enhance the manuscript’s impact.

4. Generalizability of Findings

The specificity of the Visual Short-Term Memory Binding Task (VSTMBT) is well established for early AD detection; however, the utility of nQ outside this specific paradigm remains unexplored. The manuscript would benefit from a discussion of how transferable this method might be across tasks, domains, or even other clinical populations.

5. Data Availability and Reproducibility

A portion of the neuroimaging data used in this study is not publicly available due to clinical restrictions. While this is understandable, it poses a challenge to reproducibility — particularly in a journal like PLOS ONE, which emphasizes open science. It would be helpful if the authors could at least share simulated data or a synthetic version of the dataset for validation of their pipeline.

Minor Comments

• The description of nQ is technically dense. Including an more in-depth illustrative figure or schematic could help readers better visualize the interplay of layers, modules, and nodal contributions.

• The discussion of the correlation between nQ and other network metrics (e.g., degree, PageRank, clustering coefficient) is informative; however, the manuscript could further elaborate on the implications of these relationships

Recommendation

Addressing the statistical concerns, clarifying the biological interpretation of the metric, and improving transparency around data availability will enhance the clarity and impact of this study.

6. PLOS authors have the option to publish the peer review history of their article (what does this mean?). If published, this will include your full peer review and any attached files.

**Do you want your identity to be public for this peer review?** For information about this choice, including consent withdrawal, please see our Privacy Policy.

Reviewer #1: No

Reviewer #2: No

---

## [Author Response · Author response to Decision Letter 1]

4 Jun 2025

Response:

We have double checked that the new manuscript adheres to PLOS One's style requirements. Furthermore, we have significantly edited it to remove errors (such as spelling and grammar).

ACC is funded by the Principal’s Career Development Scholarship (PCDS) for his PHD from the University of Edinburgh. This funder spent no role in the manuscript's creation/data collection. https://institute-academic-development.ed.ac.uk/postgraduate/doctoral/career-management/principals-scholarships

MAP received funding from the Alzheimer’s Society towards the Longitudinal Study of MCI through the Grants AS-R42303 and AS-SF-14-008. This funding was for the collection of MCI data (used in the manuscript) and specifically in conjunction with this research article 10.1186/s13195-022-01082-9. For the previously mentioned article, the funders only supported the study (proposed research) financially and monitored progress.

https://www.alzheimers.org.uk/

Response:

Please see the new cover letter for details.

Avalon Campbell-Cousins was supported by Edinburgh University’s Principle’s Career Development PhD Scholarship. The authors also acknowledge the support from the Alzheimer’s Society towards the Longitudinal Study of MCI through the Grants AS-R42303 and AS-SF-14-008 awarded to MAP. Additionally, the support of NHS Scotland (both Lothian and Forth Valley boards) in recruiting MCI patients and the Volunteer Panel for healthy controls at the University of Edinburgh.

ACC is funded by the Principal’s Career Development Scholarship (PCDS) for his PHD from the University of Edinburgh. This funder spent no role in the manuscript's creation/data collection. https://institute-academic-development.ed.ac.uk/postgraduate/doctoral/career-management/principals-scholarships

MAP received funding from the Alzheimer’s Society towards the Longitudinal Study of MCI through the Grants AS-R42303 and AS-SF-14-008. This funding was for the collection of MCI data (used in the manuscript) and specifically in conjunction with this research article 10.1186/s13195-022-01082-9. For the previously mentioned article, the funders only supported the study (proposed research) financially and monitored progress.

https://www.alzheimers.org.uk/

Response:

Please see the new cover letter for details.

Response:

There are legal restrictions in sharing the de-identified data set. These were imposed by the Research & Development department of the NHS Lothian Health Board - Tel: (+44) 0131 242 3330. Specifically, this data is held under their “Confidentiality and Disclosure of Information” policy as follows:

"You must not divulge Confidential Information to any third party during the period of your research or any time thereafter without the proper authority having first been given. All Confidential Information belonging to the Board, together with any copies or extracts thereof, made or acquired by you in the course of research shall be the property of the Board and must be returned to the Principal Investigator on completion of the research to which they relate or on the termination of your employment whichever is the earlier date. You will be entitled to retain any copies or extracts made or acquired by you in the course of research for references purposes only, provided that such copies or extracts are held and maintained in accordance with the provisions of the Data Protection Act 2018 and Caldicott principles."

Thus, data cannot be shared publicly without clinical research access approval from NHS Lothian and cannot be shared with any 3rd party, including extracts made from the data, given that the raw data is part of an ongoing study under the Principal Investigator Mario A. Parra. Access to some of the contents of the secondary data (current results) may be available on individual request from Javier Escudero (javier.escudero@ed.ac.uk) or Mario A. Parra (mario.parra-rodriguez@strath.ac.uk), but cannot be made publicly available due to legal constraints.

Response:

See a) for why this cannot be done.

Overview of changes (please see the PDF "Response to Reviewers" for improved formatting):

We thank the Reviewers and the PLOS One editorial team for their support and suggested changes. Here we will outline the substantial changes to the manuscript following the Reviewer suggestions:

The results and discussions sections have been heavily restructured.

Results have significantly changed following FDR correction - leading to the reduction of the number of figures and breadth of discussion. We hope that these changes promote a clearer, more robust, and concise demonstration of our results.

Several sections have been substantially condensed (introduction and method sections), reducing unnecessary details in the main text, and with additional changes in sub sectioning for clarity.

Finally, we aimed to address concerns with our communication around the limitations in sample size which we hope are now made more clear within the text, and made less worrisome with improved statistical stringency with FDR correction given that results remain strong and the main conclusions of the study do not change.

Thank you once again for your support. We believe that your comments have resulted in a stronger manuscript that will be of interest to the readership of Plos One.

Please note that tracked changes were done automatically in LaTeX. As such, some tables in the marked version of the new manuscript have poor formatting. However, note that they are properly formatted in the clean version.

Reviewer #1:

Reviewer’s Comment

In this paper, Authors propose a new metric for assessing community structure in various single-and multi-layer networks, called nodal modularity (nQ). Contrary to the more traditionally used modularity, nQ not only quantifies community structure at a global level in a network, but also provides some information about community structure at a meso-scale level (i.e. at level of single regions). NQ was tested for calculation using functional MRI (fMRI) and diffusion tensor MRI (DT MRI) data from a group of mild cognitive impairment (MCI) patients and healthy controls, calculating multiplex nQ across functional/structural layers. Results suggest that observed changes in nQ in MCI align with expected trajectories of changes in alzheirmer’s disease, suggesting applicability of nQ as a biomarker for the disease.

The paper is of interest, but it overall very long, with several redundant and unnecessary information being reported; several sections can be considerably condensed. Following major issues should be addressed.

Author’s response:

We thank you for your feedback and agree with the above assessment of the manuscript. We have aimed to address the points below with the length of the manuscript in mind. Specifically, in condensing the introduction, methods, results, and discussion sections (the final two as a result of higher statistical stringency). We hope that the new manuscript is a more acceptable length.

Reviewer’s Comment

1) The Abstract repeatedly states that nQ is very useful (compared to standard modularity) because it is a more fine-grained metric giving information not only at global level but a mesoscale level. Despite this, the results reported in the Abstract are very generic and do not describe which are the regions showing abnormal nQ. Since Authors claim that this is one of the major novelty of their study, I think that results should be expanded in this direction.

Response: We agree with the Reviewer about the need to include more detailed results in the Abstract. After restructuring the results and discussion sections, including improved statistical stringency based on Reviewer 2’s feedback, we have included more specific wording in the abstract regarding individual ROIs and their changes in nQ. Due to the high number of ROIs that were detected as statistically significant, and that many of them reside in common subsystems where deposits of amyloid-beta and tau are observed, we choose to report results in the abstract at the level of subsystems due to word count constraints. We have also shortened the abstract in several places to accommodate these changes.

Reviewer’s Comment

2) The first section of the introduction is excessively long and includes a lot of information about Alzheimer's pathology (unrelated to neuroimaging) that is mostly background and not necessary for the paper. This section should be shortened. In general, the methods section is also overly detailed and can be significantly condensed. The same applies to the number of figures and tables.

Response: We thank the Reviewer for the detailed suggestions to make the article more concise. We agree, and have removed the paragraph on Alzheimer’s pathology (2nd paragraph in the Introduction of the previous version of the manuscript). In the methods section, we have removed the description of classical modularity as it is well known, condensed the section on nodal modularity for clarity, and moved important points on modularity maximization to a separate subsection “Modularity maximization”. This makes the section shorter and better structured. Additionally, the number of figures and tables across the main text and supplemental material have been reduced following stricter statistical criteria raised by Reviewer 2.

Reviewer’s Comment

3) The end of the Introduction is not really introduction, it anticipates already results and conclusions. All the last paragraph of the introduction reporting results and conclusions should be deleted.

Response: The last paragraph of the introduction has been deleted. Thank you.

Reviewer’s Comment

4) One of the main limitations of the paper is the very small sample size of HC and MCI groups included. This should be clearly stated in the paper.

Response: We have added a statement on this to the abstract (“While limited by sample size, changes in nQ for individual regions of interest (ROIs) in our fMRI networks were predominantly observed in visual, limbic, and paralimbic systems in the brain, aligning with known AD trajectories and linked to amyloid-β and tau deposition.”) and note its presence in the “Application of nQ to investigate local scale changes in MCI “ portion of the methodology section on the choices of statistical tests (now improved with FDR, lines 399-422 in the new version of the article) and in the discussion (lines 604-605) and conclusion sections (line 651) of the paper.

Reviewer’s Comment

5) It is not clear which version of SPM software was used for fMRI analysis, some paragraphs mention SPM8 and other paragraphs mention SPM12. SPM8 is a very old software, so it is surprising that Authors use it for some analysis. To be sure that Authors are using appropriate software for the analysis, please also report software versions for Freesurfer and FSL.

Response: We are thankful for this comment. We acknowledge the two versions of SPM as an error in referencing. Specifically, in the prior version of the manuscript it states “Using SPM8 (SPM, https://www.fil.ion.ucl.ac.uk/spm/), fMRI pre-processing follows as in [34].” Ref 34 uses SPM8, and while this paper uses the same pre-processing pipeline, the version of SPM was overlooked. The current paper follows ref 34’s pre-processing pipeline but with SPM12 and the new version of the manuscript has been updated to reflect this. Furthermore, Freesurfer v5.3.0 and FSL v6.0.1 versions have now been reported on lines 146 and 148 respectively in the new version of the manuscript.

Reviewer’s Comment

6) The paragraph in the method about modularity is very long and can be condensed. The same applies for the description of the very standard network metrics (e.g. degree, clustering coefficient, centrality..). these are all very standard metrics which were described in details in hundreds of previous publications.

Response: The modularity section has been removed given that classical modularity has been well studied. Important details have been moved to a separate subsection of the methods “modularity maximization” (line 285 of the revised manuscript). The same has been done on the network metrics, only including detail on the multiplex versions of the measures. See lines 335-345 in the revised manuscript.

Reviewer’s Comment

7) I don’t understand how the fact that nQ is not correlated with the other graph metrics constitutes a "verification" of nQ. It only tells us that it is independent, but it certainly does not ensure that it is meaningful.

Response: We thank the Reviewer for this observation. We have carefully revised the text to avoid the term “verification” in this context. We now present the results in subsections “Comparison of nQ with other nodal graph measures” (line 330 of the revised manuscript) and “Independence of nQ with other nodal graph measures “ (line 449 of the revised manuscript).

Reviewer’s Comment

8) Authors should provide a Results section clearly separated from the Discussion section. In the present paper version, the t

---

## [Decision Letter · Decision Letter 1]

17 Jun 2025

PONE-D-25-03889R1Multiplex Nodal Modularity: A novel network metric for the regional analysis of amnestic mild cognitive impairment during a working memory binding task

PLOS ONE

Dear Dr. Campbell-Cousins,

Thank you for submitting your manuscript to PLOS ONE. After careful consideration, we feel that it has merit but does not fully meet PLOS ONE’s publication criteria as it currently stands. Therefore, we invite you to submit a revised version of the manuscript that addresses the points raised during the review process.

Thank you for resubmitting your work to PLOS ONE. Please make the corrections posed by Reviewer #2 so I can render a decision on this manuscript.

We look forward to receiving your revised manuscript.

Kind regards,

Stephen D. Ginsberg, Ph.D.

Section Editor

PLOS ONE

Journal Requirements:

**Comments to the Author**

1. If the authors have adequately addressed your comments raised in a previous round of review and you feel that this manuscript is now acceptable for publication, you may indicate that here to bypass the “Comments to the Author” section, enter your conflict of interest statement in the “Confidential to Editor” section, and submit your "Accept" recommendation.

Reviewer #1: All comments have been addressed

Reviewer #2: All comments have been addressed

2. Is the manuscript technically sound, and do the data support the conclusions?

Reviewer #1: Yes

Reviewer #2: Yes

3. Has the statistical analysis been performed appropriately and rigorously?

Reviewer #1: Yes

Reviewer #2: No

4. Have the authors made all data underlying the findings in their manuscript fully available?

Reviewer #1: Yes

Reviewer #2: No

5. Is the manuscript presented in an intelligible fashion and written in standard English?

Reviewer #1: Yes

Reviewer #2: Yes

6. Review Comments to the Author

Reviewer #1: (No Response)

Reviewer #2: 

Upon evaluating the revised submission alongside the authors’ detailed point-by-point responses, I am pleased to acknowledge that the majority of my major concerns have been addressed in a satisfactory and transparent manner. 

Generalizability and Broader Application

The authors have briefly addressed the applicability of the nQ metric to other paradigms and clinical populations. While the discussion could be more detailed, the current additions are appropriate given the manuscript's length and scope.

Correction for Multiple Comparisons

While the authors have correctly implemented the Benjamini–Hochberg false discovery rate (FDR) correction across all node-level statistical tests, the choice of an α threshold of 0.2 raises concerns. This threshold is notably more lenient than the conventional levels (e.g., 0.05 or 0.1) and may substantially increase the risk of false positives, potentially compromising the reliability of the findings. Although the exploratory nature and high dimensionality of the analysis are acknowledged, a threshold of 0.2 should be more thoroughly justified, including discussion of the implications for the interpretation of results. Alternatively, the authors might consider adopting a more stringent FDR threshold to balance sensitivity and specificity more effectively. Without such justification or adjustment, the statistical rigor and robustness of the conclusions remain questionable.

7. PLOS authors have the option to publish the peer review history of their article (what does this mean?). If published, this will include your full peer review and any attached files.

**Do you want your identity to be public for this peer review?** For information about this choice, including consent withdrawal, please see our Privacy Policy.

Reviewer #1: No

Reviewer #2: No

---

## [Author Response · Author response to Decision Letter 2]

2 Jul 2025

Reviewer #2:

Upon evaluating the revised submission alongside the authors’ detailed point-by-point responses, I am pleased to acknowledge that the majority of my major concerns have been addressed in a satisfactory and transparent manner.

Authors’ response:

We are grateful for your feedback and pleased to read that the manuscript has improved substantially in the revision and that the majority of your concerns have been satisfied along with all concerns of Reviewer #1. We hope that our introduction of nQ as a novel measure of nodal community structure brings a notable contribution to PLOS One.

Generalizability and Broader Application

The authors have briefly addressed the applicability of the nQ metric to other paradigms and clinical populations. While the discussion could be more detailed, the current additions are appropriate given the manuscript's length and scope.

Authors’ response:

Thank you. We are pleased that this has been addressed given the scope of the paper.

Correction for Multiple Comparisons

While the authors have correctly implemented the Benjamini–Hochberg false discovery rate (FDR) correction across all node-level statistical tests, the choice of an α threshold of 0.2 raises concerns. This threshold is notably more lenient than the conventional levels (e.g., 0.05 or 0.1) and may substantially increase the risk of false positives, potentially compromising the reliability of the findings. Although the exploratory nature and high dimensionality of the analysis are acknowledged, a threshold of 0.2 should be more thoroughly justified, including discussion of the implications for the interpretation of results. Alternatively, the authors might consider adopting a more stringent FDR threshold to balance sensitivity and specificity more effectively. Without such justification or adjustment, the statistical rigor and robustness of the conclusions remain questionable.

Author’s response:

We appreciate the Reviewer’s feedback. While 0.2 is a moderate threshold for FDR correction, this value has also been used in other imaging studies such as [1,2,3], and specifically in the context of pilot or more exploratory studies [2]. Moreover, we would like to express concern over the potential for false negatives in lowering the threshold any further. To illustrate this point, in the right panel of S1Fig(a), 25 ROIs pass FDR at 0.2, meaning that up to 20% (5 out of 25) of these ROIs may be false positives. If we were to lower the threshold to 0.1, only 3 ROIs would pass FDR correction (of which 10% of those could be false positives). This strongly suggests that 0.1 is too strict a threshold for the data given the large drop off in results, which are not proportional to the change in threshold, and thus could be inducing false negatives within the results. This is a concern especially for pilot studies such as this where limited power is at high risk for false negatives in multiple comparison correction [4]. We have already acknowledged the limited sample size of the study, which in turn means that discriminative power is driven more by effect size than p-value. Of note, FDR correction does not consider effect size or sample sizes [5].

In the previous version of the manuscript, we had stated on lines 502-504 “Note that since we are applying FDR at α = 0.2 that up to 20% of our ROIs may be false positives (equating to 5 ROIs in the multiplex model and 4 ROIs in the single-layer case)”. However, we acknowledge that this was previously limited to the results section only and not explored in the discussion. We have added the following on lines 577-578 “... (note that we focus on general changes given that up to 5 ROIs may be false positives attributed to FDR correction at α = 0.2)” in our neuroanatomical exploration of fMRI section (of the revised manuscript).

Thank you for your review and we hope that this justification and change to the manuscript addresses your concerns.

[1] 10.3389/fnhum.2013.00467

[2] https://doi.org/10.1371/journal.pone.0093623

[3] https://doi.org/10.1038/s41598-024-67177-5

[4] https://doi.org/10.1016/j.metip.2023.100120

[5] https://doi.org/10.1002/epd2.20010

---

## [Editor Report · Decision Letter 2]

6 Jul 2025

Multiplex Nodal Modularity: A novel network metric for the regional analysis of amnestic mild cognitive impairment during a working memory binding task

PONE-D-25-03889R2

Dear Dr. Campbell-Cousins,

We’re pleased to inform you that your manuscript has been judged scientifically suitable for publication and will be formally accepted for publication once it meets all outstanding technical requirements.

Kind regards,

Stephen D. Ginsberg, Ph.D.

Section Editor

PLOS ONE

---

## [Editor Report · Acceptance letter]

PONE-D-25-03889R2

PLOS ONE

Dear Dr. Campbell-Cousins,

I'm pleased to inform you that your manuscript has been deemed suitable for publication in PLOS ONE. Congratulations! Your manuscript is now being handed over to our production team.

Kind regards,

on behalf of

Dr. Stephen D. Ginsberg

Section Editor

PLOS ONE